

# Characterization of habitat requirements of European fishing spiders

Lisa Dickel[1,2,*], Jérémy Monsimet[1,*], Denis Lafage[3,4] and Olivier Devineau[1]

[1] Department of Forestry and Wildlife Management, Inland Norway University of Applied Sciences, Campus Evenstad, Koppang, Norway
[2] Centre for Biodiversity Dynamics, Department of Biology, Norwegian University of Sciences and Technology (NTNU), Trondheim, Norway
[3] Department of Environmental and Life Sciences/Biology, Karslstad University, Karlstad, Sweden
[4] CNRS, ECOBIO (Ecosystèmes, Biodiversité, Évolution)-UMR 6553, University of Rennes, Rennes, France
* These authors contributed equally to this work.

Corresponding author
Jérémy Monsimet,
jeremy.monsimet@posteo.net

## ABSTRACT

Wetlands are among the most threatened habitats in the world, and so are their species, which suffer habitat loss due to climate and land use changes. Freshwater species, and especially arthropods, receive comparatively little attention in conservation plans, and the goals to stop and reverse the destruction of wetlands published 25 years ago in a manifesto by the Union of Concerned Scientists have not been reached. In this study, we investigated the occurrence and habitat requirements at two spatial scales of two species of European fishing spiders *Dolomedes*, which rely heavily on declining wetland habitats in Sweden and southern Norway. We collected occurrence data for *Dolomedes plantarius* and *Dolomedes fimbriatus*, using a live-determination method. We modelled the placement of nursery webs to describe fine-scale habitat requirements related to vegetation and micro-climate. Using a machine learning approach, we described the habitat features for each species and for co-occurrence sites, thus providing insight into variables relevant for the presence and detectability of *Dolomedes*. Nursery placement is mostly dependent on proximity to water, presence of *Carex* sp. (Sedges) and crossing vegetation structures, and on humidity, while detection can be affected by weather conditions. Furthermore, co-occurrence sites were more similar to *D. plantarius* sites than to *D. fimbriatus* sites, whereby surrounding forest, water type and velocity, elevation and latitude were of importance for explaining which species of *Dolomedes* was present. Overall, habitat requirements were narrower for *D. plantarius* compared to *D. fimbiratus*.

## INTRODUCTION

Biodiversity is threatened by anthropogenic land use and climate change (*Sala et al., 2000*). Wetland habitats and species are declining rapidly (*Hu et al., 2017*), despite being crucial ecosystems for climate change mitigation and even human existence (*De Groot et al., 2006*). Indeed, they provide habitat for many species and are key for flood regulation and

nutrient retention (*De Groot et al., 2006*). The Ramsar Convention (*Ramsar, 2013*) and the world's scientists warning to humanity (*Kendall, 1992*) formulated wetland conservation as a global goal. However, according to *Finlayson et al. (2019)*, not only were wetland protection and restoration goals not reached, but wetlands destruction and loss have proceeded. Conservation priorities are mostly determined through variable and dynamic human values (*Lindenmayer & Hunter, 2010*), which has led to unequal conservation efforts across habitats and taxa, with groups like invertebrates (*Clark & May, 2002*; *Finlayson et al., 2019*) and freshwater/wetland species being particularly neglected (*Darwall et al., 2011*). In addition, according to a review by *Kellner & Swihart (2014)*, few studies accounted for imperfect detectability, even less so for invertebrate studies than for other taxa, which possibly affects the available knowledge about the actual status of populations and species.

Although it was recognized almost 20 years ago that there is taxonomic bias in research against arthropods (*Clark & May, 2002*), basic knowledge is still missing to inform the conservation of wetland invertebrates. This knowledge is lacking for the two European fishing spiders, namely *Dolomedes fimbriatus* and *Dolomedes plantarius*. Both species are semi-aquatic, forage on land as well as on water, and build their nursery webs close to or in vegetation above the water surface (*Gorb & Barth, 1994*; *Duffey, 2012*). The detection of both species is difficult due to their lifestyle, which includes fleeing behavior on and under the water surface when disturbed (*Gorb & Barth, 1994*). *Dolomedes* do not construct webs to capture prey, which makes individuals even more difficult to detect. But like other members of the Pisauridae family, *Dolomedes* build nursery webs (*Stratton, Suter & Miller, 2004*), which are a convenient sign of presence during the reproductive season, thus facilitating their detection. Females are found close to their nursery webs, which is useful for identification, mainly because only adults can be identified with certainty by inspecting their genitals (*Roberts, 1995*). Further, the placement of nursery webs functions as an important indicator of quality *Dolomedes* habitat since it leads to their reproductive success and survival.

Habitats of both species are declining because of anthropic transformation, including draining of wetlands (*van Helsdingen, 1993*; *Hu et al., 2017*; *Finlayson et al., 2019*). Habitats in Fennoscandia are getting more acidic due to forestry practices, resulting in an increase in acidophile plant communities (*e.g. Sphagnum, Carex*, *Blacklocke, 2016*; *Ellenberg, 1974*). While *D. fimbriatus* is relatively common (*Duffey, 2012*), *D. plantarius* is much rarer, and is one of the few red-listed spiders in Europe, despite its fairly broad distribution range (*Leroy et al., 2013*, *2014*). Naturalist observations suggest that *D. plantarius* has more specific habitat requirements than *D. fimbriatus* (*Duffey, 2012*). Acidity (low pH) has been hypothesized to be a limiting factor for *D. plantarius*' presence (*Duffey, 2012*). Habitat loss might have more severe consequences for *D. plantarius*, which has more specific habitat requirements, thus making it a species of conservation interest (*Smith, 2000*). Investigating the population decline is difficult, because historical distribution data of *Dolomedes* are scarce (*Duffey, 2012*). Some authors suggest that there may be denser populations of *D. plantarius* than known, especially in the less monitored areas in eastern Europe (in Belarus: *Ivanov, Prishepchik & Setrakova, 2017*).
Additionally, misidentifications of the two species were common in the first half of the 20th century, when body color was used for determination, although it is not a reliable indicator for the discrimination of both species (*Bonnet, 1930*; *van Helsdingen, 1993*). Little monitoring combined with potential misidentifications and difficult detection of *Dolomedes* caused an overall lack of knowledge about the distribution and status of the species. Recent observations indicate that co-occurrence, which was considered rare or even impossible, might be more frequent than previously thought (*Ivanov, Prishepchik & Setrakova, 2017*).

In this study, we contribute to further characterizing the habitat requirements of the two European *Dolomedes* species. Based on naturalist observations by *van Helsdingen (1993)*, *Duffey (1995)*, and *Duffey (2012)*, we expect *D. fimbriatus* to be more flexible than *D. plantarius* in its habitat requirements regarding the presence of water and the specific characteristics of the aquatic habitat.

## MATERIALS AND METHODS

### Study area and site choice

In order to find potential *Dolomedes* habitats, we chose our study sites based on prior observations extracted from the the Global Biodiversity Information Facility (*Global Biodiversity Information Facility, 2021*) using the R package rgbif (*Chamberlain & Boettiger, 2017*) and based on the habitat suitability map of *D. plantarius* from *Leroy et al. (2014)*. Because the resolution of the suitability map and the accuracy of the GBIF positions were too low for our purpose, we selected sampling areas within the highly suitable habitat and close to the GBIF positions based on information from the literature (*van Helsdingen, 1993*; *Duffey, 1995*, *2012*). Specifically, we chose water bodies with riparian vegetation and other types of wetlands (bogs, fens, meadows) for data collection. Because the model by *Leroy et al. (2014)* is only valid for *D. plantarius*, we assessed the potential suitability of additional sites for *D. fimbriatus* based on the visual impression we had of the wetland during a visit (applying similar criteria as for *D. plantarius*, *i.e.* proximity to water or wet habitat, but somewhat relaxed based on the predictions of *D. fimbriatus* to be less restricted compared to *D. plantarius*, *van Helsdingen, 1993*; *Duffey, 1995*, *2012*). The selected locations and the detected species are shown in Fig. 1 and more information on the sites is provided in Table S1.

### Data collection

We developed the data collection protocol by identifying relevant variables from the literature (*van Helsdingen, 1993*; *Duffey, 1995*, *2012*) followed by a pilot study. We first collected broader habitat variables at the site scale, and then we sampled multiple smaller plots along transects and around the nursery webs to reflect the microhabitat scale. We determined and recorded species of *Dolomedes* at each site (as part of site scale variables, see below). We delimited each study site by its natural borders, or, for large sites, by five transects covering 40 m along the water body (Fig. 2). Each site was visited once, and all fieldwork was carried out between the 1st of July 2018 and the 15th of August

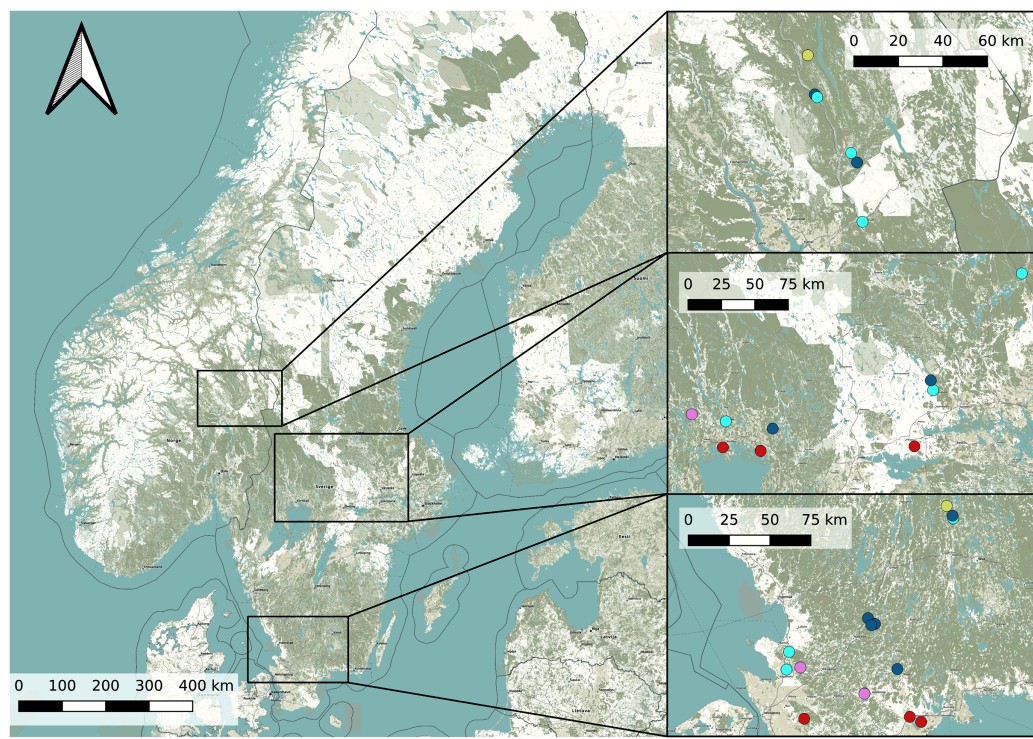

**Figure 1** **Overview map of the study area in Sweden and Norway (left) with detailed maps of the three study areas (right).** Dots represents the study sites (Pink: both species; red: *D. plantarius*; light blue: *D. fimbriatus*; dark blue: absence sites; light green: *Dolomedes* sp.). Finjasjön lake: southernmost pink dot in the bottom right map. The background map is obtained from OpenStreetMap.

2018. We collected and geo-referenced all data using the data collection software KoBoToolbox (2020; http://www.kobotoolbox.org).

### Site scale data

Since the detectability of free-ranging spiders varies with weather conditions (*Noreika et al., 2015*), we recorded temperature and wind speed, and visually classified rain and clouds at the beginning of each field work session. In case of wind (Beaufort scale >3, equivalent to 12–19 km/h wind speed) or rain, we did not attempt to detect the spiders, to keep detection conditions equal.

Two of us searched for nursery webs and spiders for 20 min before collecting data along transects. The chosen search length was long enough to cover the area, though short enough to prevent disturbing all spiders in the site due to vibration. Further, we detected spiders and nursery webs during the microhabitat data collection on transects, increasing the duration of the effective search. We searched the edge of the vegetation both visually and by sweep-netting while wading through the water if possible. If entering the water was not possible (*e.g.*, due to substrate quality, water depth, or the strength of the current), we moved carefully across the riparian vegetation to the water edge. We mostly found adult females in nursery webs or in nearby vegetation. We captured the

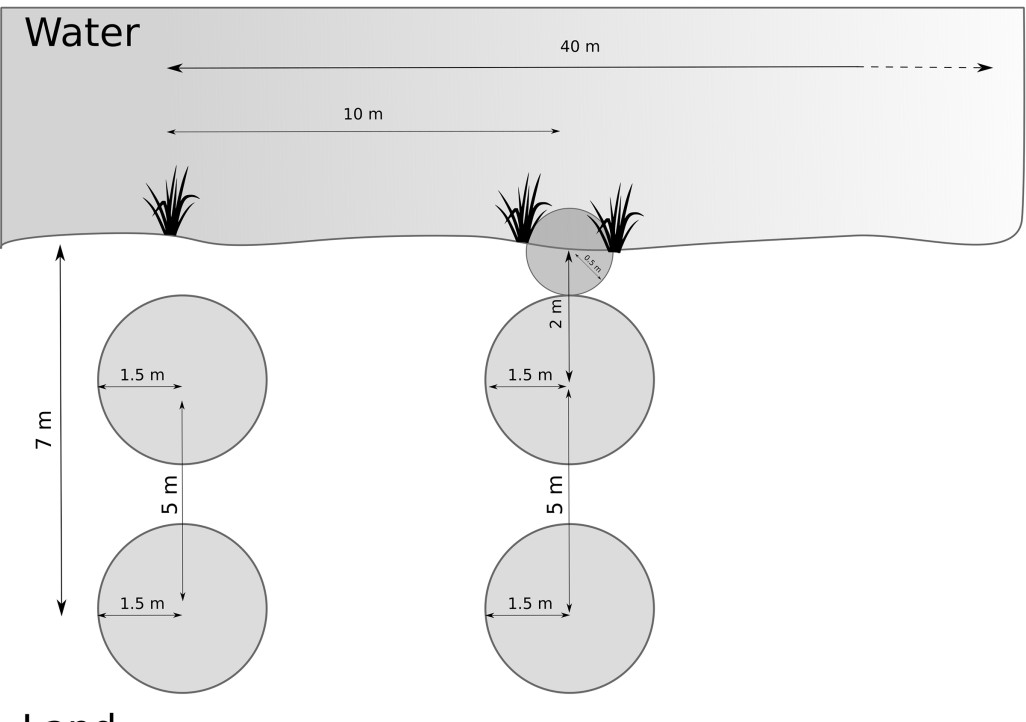

**Figure 2 Arrangement of plots and transects.** Distance between transects was 10 m, and we measured maximum 5 transects (= 40 m). Microhabitat variables and nursery presence were collected within the terrestrial plots. If the riparian vegetation or wet vegetation area was broad, data were collected as in the transect shown on the left hand. If the stripe of riparian vegetation was narrow, we inserted a half-circle directly at the water edge, as shown on the right-hand side (transect on the right).

spiders in a glass container. If the spider dived, we caught it with a fishing net (mesh size approximately 0.9 × 0.3 mm) from the water and transferred it into a glass container.

Once inside the container, we determined the species by pressing the individual gently with a soft sponge against the glass, to inspect the epigyne or pedipalps (a picture of the identification process is available in Information S2). We released all spiders after their identification. If we detected only nursery webs but no spider at a site, we discarded the data, as the nursery web of *Pisaurida mirabilis* cannot safely be distinguished from that of *Dolomedes*. When we detected and identified *Dolomedes* in a site, we assumed that all nursery webs were built by *Dolomedes*, even if we did not actually detect an adult spider near the web. We collected variables regarding vegetation type, land use, and surroundings at the site scale (Table 1). As *Dolomedes* are semi-aquatic species, measurements on microhabitat scale were concentrated around the water body or in the 'wet center' of study sites without open water, from where we drew transects for further data collection plots.

### Microhabitat data

Within each site, we collected samples at microhabitat scale to relate the specific conditions to the presence of *Dolomedes* with the goal to represent the riparian habitat in typical Scandinavian wetlands.

**Table 1 Levels and explanations of variables on site scale.**

| Variable | Levels |
| --- | --- |
| Surrounding | Infrastructure/forest/other |
| Surrounding forest | Deciduous/coniferous/mixed |
| Water type | River/bog/lake/creek |
| Water speed | Standing/slow/fast |
| Water clearness | Clear/brown/murky |
| Vegetation type (at site) | Open wet/open dry/deciduous forest/coniferous forest |
| Latitude | Latitude (continuous), Digital Elevation Model (DEM) |
| Elevation | Elevation (continuous), DEM |
| Clouds (detectability) | Yes/no/partly |
| Wind (detectability) | Measured with anemometer on Beaufort scale |
| Reason visit | Suitable habitat/GBIF/other |

We systematically arranged sampling plots along up to five transects (Fig. 2) to collect microhabitat data. If open water was present, we placed the transects perpendicular to the water body and 10 m apart. If no open water was present, we placed the transects along a wet to dry ground gradient, and if no gradient was detectable, we started the transects from a habitat edge. We recorded the applied sampling procedure for each site.

Along each transect, we collected microhabitat scale data in circular plots (radius = 1.5 m). Plots were located at 2 and 7 m from the water edge based on test sites to represent the gradient from aquatic to terrestrial habitat. The focus on the shore-area (or the wettest area in the site) is reflected by the higher density of plots close to the water (Fig. 2). When the riparian vegetation was limited to a few centimeters by the water edge, we included a half-circle ($r = 0.5$ m) plot with its center at the water edge to represent the vegetation (see Fig. 2). The shape and size of the additional plot differed from the others to avoid plots overlapping. We collected percent cover data for the five most relevant plant species according to literature (*Carex* spp., *Juncus* spp., *Typha* spp., *Phragmites* spp. and *Sphagnum* spp.) on the Braun-Blanquet scale (*Westhoff & Van Der Maarel, 1978*), which we later simplified for modeling purposes (Table 2). Some of these species, including *Carex* spp. and *Sphagnum* spp., thrive in low pH habitats, so this variable also correlates to the microhabitat feature of low pH. Furthermore, we collected structural and microclimate variables (described in Table 3). We collected the same measurements around the nursery webs, which we searched for in the entire site. We extracted site elevation after data collection from a digital elevation model (*EEA, 2018*).

## Statistical analyses

We prepared and analyzed all data in R (*R Core Team, 2020*), and R Studio (*RStudio Team, 2012*). We followed the protocol for data exploration by *Zuur, Ieno & Elphick (2010)* and used the tidyverse framework for data exploration and preparation (*Wickham et al., 2019*). We standardized all continuous variables to facilitate model convergence and interpretation.

**Table 2 Braun Blanquet scale and simplification used in this study.** The Braun Blanquet scale combines both cover and abundance (percentage plant coverage and number of occurrences). The simplified categories were used to reduce the degree of freedom of our models (Table adapted from *Westhoff & Van Der Maarel, 1978*).

| Percentage plant coverage | Number of occurrences | Original Braun Blanquet category | Simplified category |
|---|---|---|---|
| 0 | 0 | no | 0 |
| <5 | 1 | r | |
| <5 | 2–5 | + | |
| <5 | Abundant | 1 | 1 |
| 5–25 | Very abundant | 2 | |
| 26–50 | Arbitrary | 3 | 2 |
| 51–75 | Arbitrary | 4 | 3 |
| 76–100 | Arbitrary | 5 | 4 |

**Table 3 Levels and explanations of variables on microhabitat scale.** BB: Braun Blanquet.

| Variable | Levels |
|---|---|
| Spiders detection | Spider/nursery web/no |
| Distance to water | No water; 0 m; 0.7 m; 2 m; 7 m |
| Humidity | Measured at ground level and 20 cm above ground |
| Horizontal cover | Visually assessed at 10 cm; 30 cm; 50 cm above ground, indicating the proportion of the area covered by vegetation |
| Maximum height | Maximal height of vegetation in plot, measured with 10 cm accuracy |
| Average height | Measured 5 times in random location within plot with 10 cm accuracy |
| Tussocs | Tuft of grasses or sedges, measured on BB scale |
| Large leaves | Yes/no; this variable indicates green plant parts which are deciduous or broad (opposed to stems or grass shaped leaves) |
| Litter | Yes/no |
| Shade | Yes/no/partly |
| Crossing structures | When stems or leaves cross to form a "platform", measured on BB scale |
| *Carex* spp. | Cover-abundance measured on BB scale |
| *Juncus* spp. | Cover-abundance measured on BB scale |
| *Typha* spp. | Cover-abundance measured on BB scale |
| *Phragmites* spp. | Cover-abundance measured on BB scale |
| *Sphagnum* spp. | Cover-abundance measured on BB scale |
| Deciduous plants | Cover-abundance measured on BB scale |
| Aquatic vegetation | Yes/no |
| Nursery web detected | Yes/no |

### Site scale analysis

In order to investigate differences among occupied habitats, we compared sites in which only *D. fimbriatus*, only *D. plantarius*, both species, or neither species were detected by using flexible discriminant analysis (FDA; *Hastie, Tibshirani & Friedman, 2009*; *Hastie, Tibshirani & Buja, 1994*) using the R package 'mda' (*Hastie et al., 2013*). We used the

*Dolomedes* species detection (*i.e. D. fimbriatus, D. plantarius*, both species, or neither species detected) as the response variable. We considered surrounding landscape and forest type, latitude, elevation, water type, water speed, water clearness, and vegetation type as predictors (Table 1).

In addition, we used a single-season occupancy model (*MacKenzie et al., 2002*) within the unmarked package (*Fiske & Chandler, 2011*) to predict the nursery detection probability pooled for both species (because Dolomedes species data are only available at the site scale). We used the microhabitat scale data (collected around nursery webs and along transects) as spatial replicates, as an alternative to the usual temporal replicates/detection attempts. We considered weather, microclimatic variables (wind, cloudiness, rain, shade), vegetation structure, and sampling related variables (see Table 1) as potentially influencing detectability.

### Microhabitat characteristics around nursery

We modeled nursery presence/absence for sites where we verified the presence of *Dolomedes* and found at least one nursery web. Therefore, we ensured that the sampling was not temporally unsuitable or the site generally unsuitable, which allowed us to model nursery placement within generally suitable sites.

For variable selection and parameter estimation, we fitted a binomial Generalized Additive Model (GAM) by component-wise boosting, using package mboost (function gamboost, *Hothorn et al., 2021*). Prior to model fitting, we checked variables correlation and dropped highly correlated variables (threshold = 0.7, *Dormann et al., 2013*): we retained humidity at ground level and average vegetation height and removed humidity at 20 cm and maximum vegetation height.

We did not consider interactions due to the low sample size. We then fitted a regularized model, following the recommendation in *Hofner et al. (2018)* using all other predictor variables to identify the most relevant predictors and estimate the model parameters. We validated the model using cross validation and present the final model estimates.

To validate the model, we tested the stability of the selected variables *via* resampling using the package 'stabs' (*Hofner & Hothorn, 2017*). Stability selection provides a reliable way to find an appropriate level of regularization, to keep variables with high selection probabilities. In our model, we used standard choices of tuning parameters, with a cut-off of 0.75 and the number of falsely selected base learners tolerated of 1 (*Meinshausen & Bühlmann, 2010*).

## RESULTS

### Site scale habitat characteristics–species specific

We detected *D. fimbriatus* alone in 12 sites, *D. plantarius* alone in 6 sites, both species together in 4 sites and none of the two species in 9 of the visited sites (total sites: $n = 31$, Fig. 1).

The first axis of the FDA explained 77.3% of the variation, the second axis 14.1% of the variation (Fig. 3). The main variables loading onto the FDA axes, *i.e.*, separating best

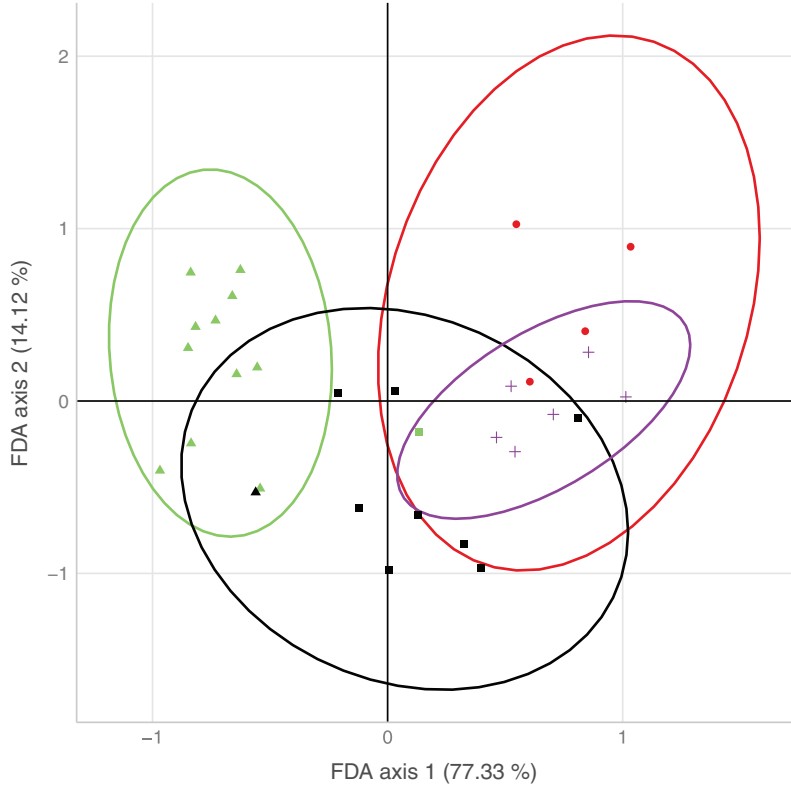

**Figure 3 Plot of the results from the flexible discriminant analysis (FDA), where colors represent observed occurrences of the species, shapes predicted occurrences, and ellipses indicate uncertainty of predicted species occurrences (95% confidence intervals).** Presence of both species (observed: red, predicted: circle), only *D. fimbriatus* (observed: green, predicted: triangle), only *D. plantarius* (observed: purple, predicted: cross) and no *Dolomedes* (observed: black, predicted: square).

between sites with none/both/each species, were water type and surrounding forest on the first axis, and water speed on the second axis.

Sites holding either of the two species were well separated by the combination of variables loading on the first axis. Sites with only *D. plantarius* were more restricted with respect to the associated habitat variables, *i.e.* water type, surrounding forest and water speed as the top loading variables, and sites with both species overlapped mostly with *D. plantarius*-only sites (Fig. 3, Table 4).

## Microhabitat characteristics around nursery

In the field, we found nurseries in 35 plots out of 184. The main variables selected in the boosted GAM model (Fig. 4) were distance to water (variable importance = 67.3%), *Carex* spp. cover (variable importance = 11.7%), crossing structures (variable importance = 1.1%), the random effect site ID (variable importance = 1.3%), humidity at ground level (variable importance = 3.6%) and the intercept (variable importance = 15.1%). We found that high abundances of sedges (*Carex* spp.), crossing structures, high values of humidity and low distances to water increased the probability of the presence of a *Dolomedes* nursery (Fig. 5). If water was present, the probability of encountering nursery

**Table 4 Loading values of the variables in the three axes of the FDA (flexible discriminant analysis).**

| Variable | Category | Axis 1 | Axis 2 | Axis 3 |
|---|---|---|---|---|
| Elevation | | −0.3 | −0.4 | −0.54 |
| Latitude | | −0.22 | 0.22 | −0.13 |
| Surrounding and forest | Pine deciduous | −0.27 | −0.32 | −0.21 |
| | Other | −0.27 | 0.26 | 0.57 |
| | Pine | −0.23 | 0.24 | 0.06 |
| | Deciduous | −0.1 | 0.37 | −0.69 |
| | Spruce | 0.02 | −0.46 | 0.08 |
| | Infrastructure | 0.09 | −0.13 | −0.12 |
| | Fields | 0.77 | 0.04 | 0.32 |
| Vegetation type | Open dry | −0.69 | −0.06 | −0.15 |
| | Open wet | −0.04 | 0.2 | −0.22 |
| | Coniferous | 0.34 | 0.09 | −0.03 |
| | Deciduous | 0.37 | 0.23 | 0.32 |
| Water clearness | Clear | −0.15 | −0.38 | 0.5 |
| | No water | −0.05 | 0.17 | −0.15 |
| | Murky | −0.04 | −0.76 | −0.03 |
| Water speed | No water | −0.05 | 0.17 | −0.15 |
| | Slow | 0.04 | −0.07 | 0.51 |
| | Standing | 0.47 | −0.01 | −0.14 |
| Water type | Other | 0.16 | 0.02 | −0.17 |
| | Creek | 0.29 | 0.56 | −0.61 |
| | Lake | 0.39 | −0.02 | 0.18 |
| | River | 0.55 | −0.51 | −0.27 |

webs beyond 70 cm away from the water edge was low. There was variation in the probability of finding nursery webs across sites (Fig. 5). However, when testing the stability of the selected variables *via* resampling, only distance to water, the intercept and the random site ID were found to be stable enough, which was most likely caused by the small sample size.

The detection probability of nursery webs was higher for plots with a high abundance of crossing structures, higher air temperatures, fewer clouds (at the time of data collection), as well as for sites with open water compared to sites without a water body. Model details can be found in Table S2.

## DISCUSSION

In this study, we found that *D. fimbriatus* is more of a habitat-generalist than *D. plantarius*. The habitat requirements of the two species were discriminated by forested habitats, and habitats with low pH (as indicated by the presence of species such as *Carex* spp. and *Sphagnum* spp., *Ellenberg, 1974*, and the proximity to coniferous forest, *Blacklocke, 2016*) or absence of water. We frequently found *D. fimbriatus* in forested areas, especially in coniferous forest, while we never encountered *D. plantarius* in these habitats. *Duffey (1995)*

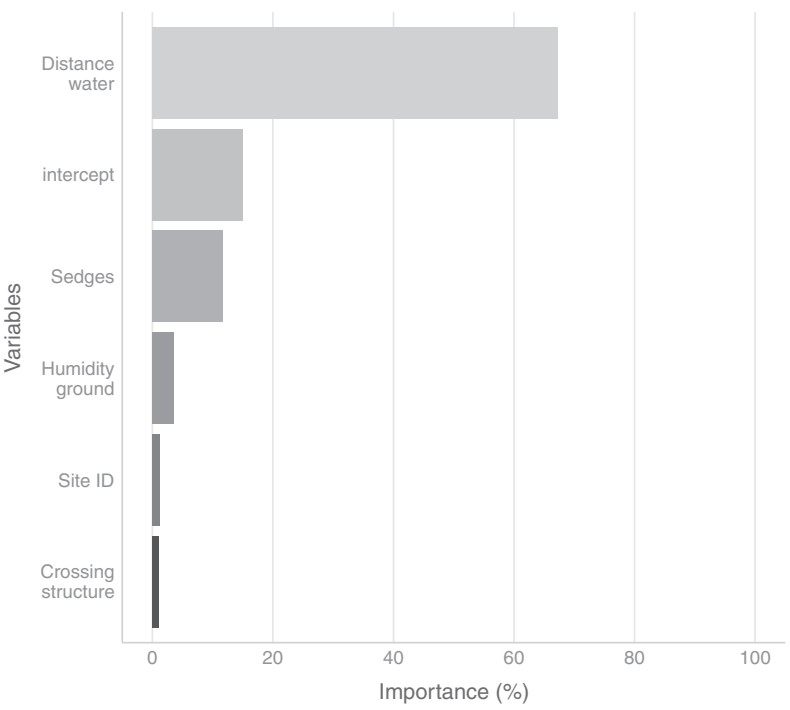

**Figure 4 Variables importance for the nursery placement model (boosted GAM).** The "intercept" variable is the combination of the first level of each factor at humidity = 0. Distance water: distance to water, Sedges: abundance of *Carex* spp. on simplified Braun-Blanquet scale, Crossing structure: crossing vegetation structures on simplified Braun-Blanquet-scale, Site ID: varying intercept per site ID, Humidity ground: humidity at ground level (standardized).

hypothesized that *D. fimbriatus* can occupy habitat with lower pH values compared to *D. plantarius*. Surrounding coniferous forests, which are dominant in Fennoscandia, may acidify water streams (*Blacklocke, 2016*), thus impacting the pH and potentially restricting *D. plantarius*. We found *D. plantarius* most often at sites with slow-flowing rivers, and we found *D. fimbriatus* most often in bogs. *D. plantarius* was also highly associated with open and slow water, whereas *D. fimbriatus* was less restricted by water conditions. *Dolomedes* can use water as a hunting area and benefit from the use of vibrations at the water surface to detect prey (*Bleckmann & Lotz, 1987*). This close relationship to water, together with the observation of juveniles of *D. fimbriatus* far from the shore, while juveniles of *D. plantarius* are found on the water (*Duffey, 2012*), might result from different hunting abilities of the two fishing spider species.

We found some overlap in habitat requirements, which reflected a spatio-temporal overlap at the site scale (see Fig. 3). *Holec (2000)* hypothesized that co-occurrence of both species might only be observed in transitional habitats between sites suitable for *D. plantarius* (*i.e.* ponds) and sites suitable for *D. fimbriatus* (*i.e.* bogs). This observation is validated for one of the sympatric sites sampled (Fig. 1), a fen in the forest. Nonetheless, we hypothesize that the conditions for co-occurrence are less restrictive because similar to *Lecigne (2016)*, we found two sympatric populations on the vegetation at the shore of a lake (Finjasjön lake, Fig. 1). As already hypothesized by *Duffey (2012)* and *Duffey (1995)*, and *van Helsdingen (1993)*, our data confirmed the higher degree of association of

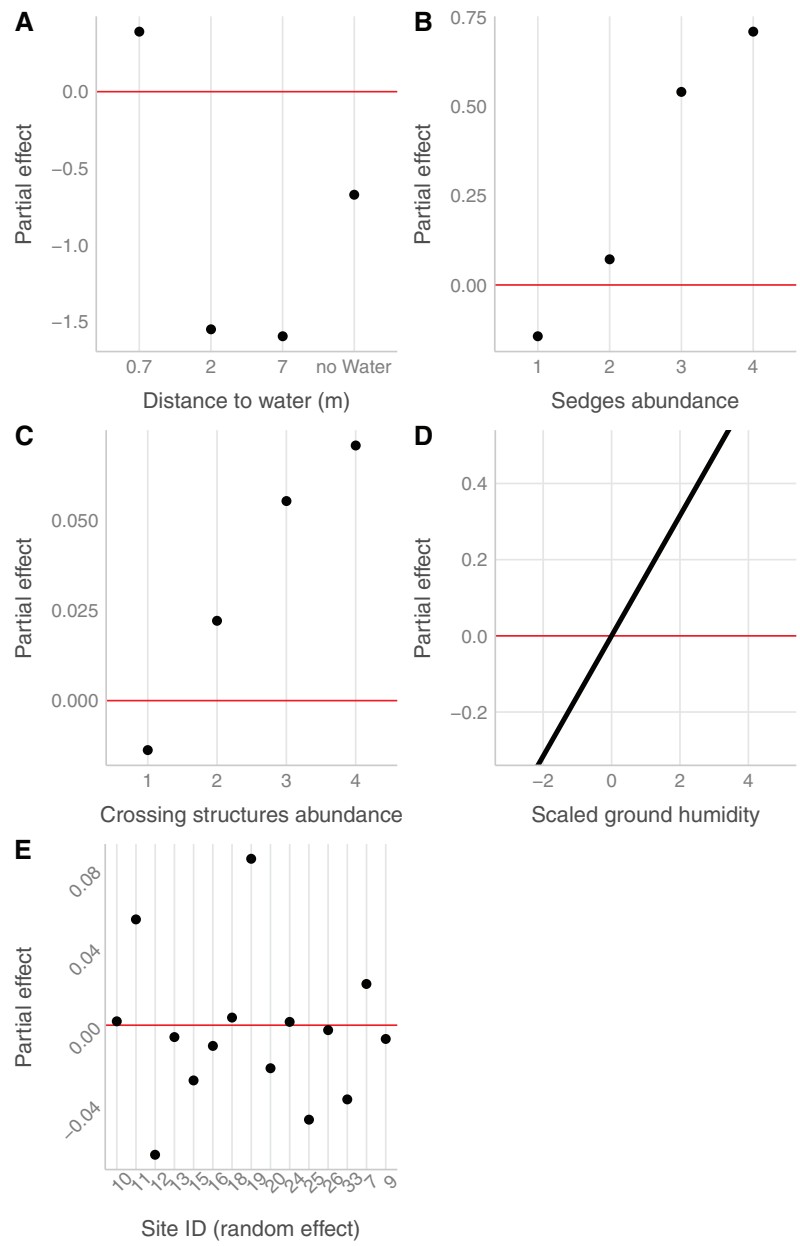

**Figure 5 Marginal effects of variables selected in the nursery placement model (boosted GAM).** Value above/below zero represents a positive/negative effect, with the further the value is from the red line, the greater is the effect. (A) Impact of the distance to water (categorical variable, category 0 meter in the intercept), (B) of the abundance of sedges (categorical variable, intercept: 0), (C) of the abundance of crossing vegetation structures (categorical variable, intercept: 0) and (D) of the ground humidity (continuous variable) on the presence of nursery. (E) The effect of sites (categorical variable), considered as random effect.

*D. plantarius* with water compared to *D. fimbriatus*, and in general more substantial restrictions in potential habitats for the former. This suggests that *D. plantarius* is more specialized in its habitat requirements than *D. fimbriatus*. The co-occurrence of the two species might be explained by a broader ecological niche of *D. fimbriatus*, which partly

overlaps with the niche of *D. plantarius*. The co-occurrence observed hides a possible segregation of both fishing spider species at the microhabitat scale.

At the microhabitat scale, *D. plantarius* might be more dependent on water for its reproductive behavior and nursery placement, which will require further species-specific investigation. Moreover, distance to water and humidity of the ground influenced nursery web placement. This dependency of *D. plantarius* on the water could facilitate cohabitation with *D. plantarius* being spatially segregated towards the shore. We also observed, in two sympatric populations, *D. plantarius* females carrying egg-sac while females of *D. fimbriatus* were already guarding their nursery webs with spiderlings. This might indicate temporal segregation as well, which would also facilitate the co-occurrence of otherwise ecologically close spider species (*Uetz, 1977*; *Fasola & Mogavero, 1995*). Lastly, a segregation for food could occur based on the different diet of the two species, with juveniles of *D. plantarius* being more restricted to water (*Duffey, 2012*).

Within habitats occupied by *Dolomedes*, we found at the microhabitat scale that abundance of sedges (*Carex* sp.) and crossing structures, together with distance to water and humidity, were the most relevant variables for predicting the presence of nursery webs. Indeed, the architecture complexity of the vegetation, as well as the relation between plant community and architecture, are important for wandering spiders (*Woodcock et al., 2007*; *Vasconcellos-Neto et al., 2017*). Here, this is expressed by the positive influence of the presence of crossing structures. We also hypothesize that spiders benefit from the stiff stems of the sedge more than being taxonomically exclusive to them for placing nurseries. *De Omena & Romero (2008)* showed that some species which are associated with specific host plants are sometimes mostly dependent on the plant's architectural structure for hunting and dwelling. Specifically, prey abundance and resulting prey-predator interactions can depend on vegetation structure (*Denno, Finke & Langellotto, 2005*). Thereby, there are multiple possible explanations for the importance of vegetation structure for *Dolomedes*.

In this study, our sample size was small due to the rarity of the two species, especially of *D. plantarius*, and to a narrow temporal window for data collection. At the site scale, this small sample, and especially the lack of co-occurrence sites limits the scope of our conclusions about the characteristics of sympatric populations. At the microhabitat scale, repeated visits of the same sites would provide opportunities to refine the occupancy model and to clarify detection issues for these two species. With a better knowledge of nursery timing, other microhabitat studies would also be facilitated. Further data collection at the landscape level would increase knowledge about potential habitat, and investigating water and soil acidity could be helpful to clarify habitat restrictions for *D. plantarius*. Finally, species-specific occupancy modeling could be helpful, especially because *D. plantarius* is likely to dive when disturbed and might be more difficult to detect than *D. fimbriatus*, which might prevent identification of double-species sites.

The habitat suitability for both species is expected to shift northward in Europe in response to climate change (*Leroy et al., 2013*, *2014*; *Monsimet et al., 2020*). This shift might be limited by low dispersal abilities and unconnected habitats in Fennoscandia

(*Monsimet et al., 2020*). The lower dispersal abilities of *D. plantarius* (J. Monsimet, 2021, unpublished data), combined with its narrower habitat requirements may explain its more restricted distribution and scarcer populations. It is therefore essential to protect both current and future habitats. Conserving both *Dolomedes* species emphasizes the special importance of protecting wetlands, in Fennoscandia and elsewhere (*Sala et al., 2000*; *Davidson, 2014*; *Carson et al., 2019*). The conservation of the red-listed *D. plantarius* might be prioritized as it seems to have narrower habitat requirements than *D. fimbriatus*, which makes it more vulnerable to climate change (*Cardoso et al., 2020*).

To counteract various threats, which spiders currently face, land protection and the management of both land and species, is important (*Branco & Cardoso, 2020*). For efficient management, estimating the local probability of presence of the species is also important. Occupancy modeling can help to decide which areas could be necessary to protect and where to apply conservation efforts (*McFarland et al., 2012*). In this study, the detection probability of nursery webs was higher where abundance of crossing vegetation structures was high and with good weather condition, *i.e.*, optimal temperature and sunny weather. Nonetheless, the use of nursery webs as detection units could be improved by specifying the timing and duration of nursery webs with repeated visits (*e.g.* weekly) to the same sites and nursery webs (*Smith, 2000*). Monitoring nursery webs also makes it possible to encounter the female spiders, which is especially valuable with the non-invasive sponge-technique we used for identifying the species. Furthermore, spatial patterns between adults and webs of the same and different species of *Dolomedes* could give further insight into co-occurrence as well as abundance. Spatial patterns, beside habitat characteristics, could potentially arise due to positive (*e.g.* finding a mate) or negative interactions (*e.g.* cannibalism, predation) between individuals of the same or different *Dolomedes* species or just due to the lack of dispersal of individuals. In addition to estimating the population's abundance dynamics, preserving shorelines with abundant crossing structures and continuously web habitats are essential to managing and conserving populations of *D. plantarius*.

## CONCLUSIONS

In this study, we found that *D. fimbriatus* is more of a habitat-generalist than *D. plantarius*. The former can occupy sites with a lower pH, indicated here by the presence of characteristic vegetation and the proximity to coniferous forest. *D. plantarius* is also less tolerant to the absence of water. Moreover, we found some overlap in their habitat, with the overlap site more similar to *D. plantarius* sites.

Abundance of sedges, crossing structures, distance to water and ground humidity influence the presence of nursery webs. The information at the site and the microhabitat scales provides relevant information for the management of both species.

## ACKNOWLEDGEMENTS

We thank Lars Jonsson (Kristianstad University) for his advice and help during the data collection, Andres Ordiz for comments on a previous manuscript version, and Boris Leroy for sharing with us his prediction maps to select sites.

We would like to thank Marc Milne and two anonymous referees who provided useful and detailed comments on the manuscript.

### Funding

This work was supported by Inland Norway University of Applied Sciences through a doctoral scholarship granted to Jérémy Monsimet from 2018 to 2021. The funders had no role in study design, data collection and analysis, decision to publish, or preparation of the manuscript.

### Grant Disclosures

The following grant information was disclosed by the authors:
Inland Norway University of Applied Sciences.

### Competing Interests

The authors declare that they have no competing interests.

### Author Contributions

- Lisa Dickel conceived and designed the experiments, performed the experiments, analyzed the data, prepared figures and/or tables, authored or reviewed drafts of the paper, and approved the final draft.
- Jérémy Monsimet conceived and designed the experiments, performed the experiments, prepared figures and/or tables, authored or reviewed drafts of the paper, and approved the final draft.
- Denis Lafage conceived and designed the experiments, authored or reviewed drafts of the paper, and approved the final draft.
- Olivier Devineau conceived and designed the experiments, analyzed the data, authored or reviewed drafts of the paper, and approved the final draft.

### Field Study Permissions

The following information was supplied relating to field study approvals (*i.e.*, approving body and any reference numbers):
Not applicable.

### Data Availability

The data and R scripts of the statistical analyses are available at GitLab: https://gitlab.com/Monsimet/HabRequirementsFishingSpiders.

### Supplemental Information

Supplemental information for this article can be found online at http://dx.doi.org/10.7717/peerj.12806#supplemental-information.

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
