# Peer review of "Characterization of habitat requirements of European fishing spiders"

_PeerJ, doi:10.7717/peerj.12806_

## Round 0.1 · original submission · Major Revisions

Dear Monsimet et al.,

In light of the three reviews we received regarding the evaluation of your study, I consider that it may be published in PeerJ after the text is reviewed, considering the improvement suggestions provided. I am confident you will be able to perform all the necessary changes.

Sincerly, Daniel Silva.

·

Basic reporting

Basic Reporting
• The English in this manuscript is good, but not perfect. I made some suggestions for improvements here:
o Lines 30-32: I’m not sure why the statement about habitat requirements is put together here with the statement about detecting nursery webs. It seems like these two things are not related. The first half of this sentence basically states that habitat requirements are narrower for one species than another. Shouldn’t the second half (after the semicolon) then talk about those habitat requirements? Instead, it discusses how detection of webs is affected by the weather. It mostly seems like the second half of this sentence is more related to the next sentence.
o Lines 46-47: “not only were wetland protection and restoration goals not reached, but”
o Lines 51-53: This last sentence of your introductory paragraph is confusing. What do you mean by “imperfect detection”? Detection of invertebrates? Perhaps you need another sentence before this one that links poor conservation work on invertebrates to their inability to be noticed or seen or detected.
o Line 58: “close to or high above the”
o Lines 67-68: “an important indicator of quality Dolomedes habitat since it has led to its reproductive success and survival.”
o Line 110: I’m not sure what you mean by “at the site scale.” Perhaps use “recorded species of Dolomedes while collecting site scale data.”?
o Line 116: Ensure that there is a space after 12 and before your hyphen.
o Lines 120-121: “cover the area, though short enough”
o Line 122: “nursery webs”
o Lines 127-128: “females in nursery webs, in nearby vegetation, or on the water.”
o Line 132: use a lowercase “a” in beginning your parenthetical phrase.
o Line 134: “but no spider at a site, we”
o Line 135: “from that of Dolomedes.”
o Line 136: “vegetation type, land use, and surroundings at the site scale (Table 1).”
o Lines 168-170: “only D. fimbriatus, only D. plantarius, both species, or neither species were detected” same for line 172.
o Line 180: “vegetation structure, and sampling related”
o Lines 191-192: I do not understand this sentence: “From variables humidity at ground level…we kept humidity at ground level.” Are you saying that you discarded one of those variables and kept the other like vegetation height in the next sentence? Is this because you dropped highly correlated variables as you mention previously? If so, you should state these variables in or right after that sentence. For example, “…we dropped highly correlated variables: humidity at 20cm above ground level was dropped for humidity at ground level and maximum vegetation height was dropped for average vegetation height.” Or, if you decide to make it multiple sentences, keep them together. Currently you have a statement about variable interaction (Line 190-191: “We did not consider interactions due to the low sample size.”) between these two related sentences and it makes it confusing.
o Line 207: explained that percent of the variation, right? If so, say that.
o Line 210: do you mean well defined rather than “well discriminated”?
o Line 211: do you mean D. plantarius-only sites?
o Line 213: “sites with both species
o Line 221: “of the presence of”
o Line 233: “is more of a habitat-generalist”
o Line 237: “tolerant” implies that it is fine with it there but doesn’t particularly enjoy its presence. Perhaps you mean “favored” as in “D. fimbriatus favored forested areas”?
o Paragraph between 249-262: I agree with your assessment here. You should also reference Figure 3 here as it clearly shows that the habitats where co-occurrence happens fully overlaps (within 95% CI) the habitats of D. plantarius but not D. fimbriatus, thereby giving a greater number of habitat variables for the latter compared to the former.
o Line 305: “presence of the species is also important.”
o Line 307: “the existence of nursery webs was”
o Line 318: “due to the lack of dispersal”
o Line 318-321: This last sentence here is confusing. Perhaps, “Preserving shorelines with abundant crossing structures and continuously wet habitats are essential to managing and conserving populations of D. plantarius.”?
o Line 324: “more of a habitat-generalist”
o Line 325: “can occupy sites with a lower pH, indicated”
o Line 359 and 370: why did you just say “others” for the authors on these manuscripts when you included all of the authors for the Cardoso paper, which has more?
• The authors provide sufficient references, but I wish their introduction included more information relevant to their conclusions:
o Specifically, they should include information on the role of pH in wetlands in Fennoscandia, specifically low pH, and its relation to vegetation (what species are present) and spiders.
o Lines 74-75: Does D. plantarius have less plasticity? Is this evidenced in Smith, 2000? It seems that Smith, 2000 only discusses this species and does not compare it to D. fibriatus.
• Article structure is sufficient, but I could not access some of their raw data. Specifically, Nursery_data.csv is not a .csv file as it states and is instead “nursery_data(1).gz” and I was not able to open it. Moreover, some figures and tables are problematic in that they have vague captioning, unclear terminology, and errors. I have included the specifics here.
o Figure 1: Your study site colors in the caption do not match the colors in the figure. Your caption state pink, red, green, blue, and light green. In the figure, I see pink, red, light blue, dark blue, and yellow (which, admittedly, may be light green). Also, for your caption, you should call them “detailed maps” rather than “detail maps.” Lastly, it seems like you’re missing a few periods in the last part of the caption.
o Figure 2: This figure is not the easiest to understand. It helped when I realized that the top middle plot was enlarged in the bottom right, but that took a while to determine. Second, when you say “aquatic plot” it seems that part of these plots border land? For example, looking at the top right plot, it’s half in the water and half on the land but it’s labeled “w” as an aquatic plot. Was the land part of that circle not sampled?
o Figure 3: The “both species” variable is “red, circle” not “red, square”.
o Figure 4: The figure by itself seems fine but it is confusing that the numbers in the figure do not correspond or correlate to the figures given in the sentence in the text where this figure is cited. In the text, you cite “selection frequency” values and in the figure you cite “importance (%)” on the y-axis. These are clearly not the same and the values do not match because of this. Could you use the same values for both text and figure to improve clarity? You also define the first variable in the caption: “distanceWater” as distance to water. Then you define the second variable “rCarex” but that’s not listed as a variable in the figure. “Sedges” is – is this the same thing? I’m also assuming that “rCrossingStr” is Crossing Structure? These need to be clearly linked.
o Figure 5: The caption needs to do a better job at explaining this figure. What does the red line represent? Why is figure 5D a line while all the others are discrete points?
o Table 1: Do you use “landscape scale” and “site scale” to mean the same thing? If so, please use one or the other.
o Table 2: This table is confusing. Use “Percent plant coverage” instead of “percentage.” Explain in the capture which plants are considered as you did in the text. Why are there periods between “Number of occurrences” instead of spaces? It looks like you also collected occurrence data for these plants, though you don’t say so in the text. I’m assuming this is number of plots that it occurred in? It’s unclear, though. What do you mean by “arbitrary”?
o Table 3: Do you use “microhabitat scale” and “nursery scale” to mean the same thing? If so, please use one or the other. Your first variable lists “spider/nursery web/no” as your levels. However, you explain on lines 133-136 that if you ever saw just a nursery web, you would discard the data “as the nursery web of Pisaurina mirabilis cannot safely be distinguished from the web of Dolomedes.” Therefore, none of your data should have “nursery web” as a variable unless it also includes spider as well, right? I’m assuming this also applies to all of the nursery web data at the bottom of this table? You have distance to water as 0m, 0.7m, 2m, and 7m. Where in the text do you describe the 0.7m plot? Why aren’t you including the 12m plot that you described in the text? There are many variables here, but many are vague with no description in the text. For example, what constituted “large leaves”? How was horizontal cover measured? How was humidity measured? None of these things are described in the figure caption or text.
• The article is self-contained and all results are relevant to their hypotheses.

Experimental design

Experimental Design
• The research is original primary research and does fit within the Aims and Scope of PeerJ.
• The research question is well defined, relevant, and meaningful, and it is state how the research fills an identified knowledge gap.
• The investigation has limitations, though the authors admit these limitations in their discussion. Specifically, they admit that their sample size is small and that their sampling was limited, preventing them from making some more advanced conclusions.
• The methods need some work. It is vague at times and missing relevant and important information. I have included my comments here:
o Methods: I can tell by your Supplemental Table 1 that you collected data daily over a month and a half period, but I don’t see where you say that in the text. Nor do you say that you sampled largely in July and August.
o Lines 97-100: When you say, “based on information from the literature”, you are referring to the next paragraph where you state the habitats that van Helsdingen and Duffey reference? If so, I would make this entire section one paragraph and start the “We chose water bodies” sentence with, “Specifically, we chose water bodies…”
o Lines 104: We need more information for D. fimbriatus habitat other than “the visual impression we had of the wetland during a visit.” What specific habitat features were you looking for? How did these features differ from the D. plantarius habitat? Why were you looking for those features? Specifically, which articles reference those features for D. fimbriatus?
o Line 113: I think it would help if you first stated what “Site scale data” are before you got into the specifics of how collection was done. Same for “Plot scale data”, which you call “microhabitat data.”
o It’s confusing that you have so many terms for your data. For example, at some points you use “site scale data”, “systematic data”, “microhabitat data”, “plot-scale data”, “percent cover data”, and “nursery data.” Do any of these overlap? Is it possible to maybe group some of these or use the same terms so that it’s less confusing as to which data you’re referring to? This goes for scale as well. Oftentimes you use “level” and “scale” interchangeably. Use just one of these two to reduce confusion.
o Line 144: You say that sometimes transects were started at random. How was randomness decided – through a random number generator? Was this truly random or haphazardly chosen?
o Lines 148-150: Terrestrial plots were located at 2, 7, and 12 meters from the water edge but Fig. 2 doesn’t show this? Fig. 2 shows that 7m is the farthest distance. Why the discrepancy?

Validity of the findings

Validity of the Findings
• Findings are novel and add to our knowledge of Dolomedes habitat preferences within Fennoscandia. Conclusions are related back to the conservation status of one of the two species studied.
• Data have been provided. The data I could access seem sound.
• Results and conclusions are well stated but get confusing in some areas, especially when referring to the different variable that had effects on the habitat estimations. Some specific examples are here:
o Lines 227-229: You cite Figure 3 here, but I thought you explained Figure 3 earlier – in the first paragraph - as being largely influenced by water type, surrounding forest, and water speed. So, why are you now citing this figure and talking about different variables (higher air temperature, fewer clouds, etc.)?
o Line 235: One of your main discussion points and conclusion points is that a low pH helped discriminate sites. However, you never mention pH until the discussion. Include some background about pH, plants that thrive in low pH, and habitats with low pH in your introduction. You may also want to mention the association between those wetland plants and pH in your methods and results (e.g., “We collected percent cover data for five of the most relevant plant species according to the literature (Carex spp., Juncus…etc.) on the Braun-Blanquet scale. Many of these species, including X, Y, and Z, thrive in low pH habitats, so this variable also correlates to the microhabitat feature of low pH.” In fact, the information in this paragraph about Duffey’s hypotheses and how coniferous forests are dominant in Fennoscandia may better be placed in the introduction as features that may restrict Dolomedes ranges.

Additional comments

General comments
• Authors also provide interesting observations of temporal segregation, especially when combined with their microhabitat data.

Reviewer 2 ·

Basic reporting

A few careless mistakes were found in the way scientific names were written.
For example,
L 87 The genus name should be italicized
L156 spp. should not be italicized.
There are no other major problems. English writing, logical structure, and references are not a problem.

Experimental design

Site Scale Data: As the authors are working at a large spatial scale, it was felt necessary to take latitude (or climatic factors) into account in the analysis. In fact, it might be difficult to take them into account because of the small sample size, but it would have been better to mention their influence.

Regarding the microhabitat survey, I wonder what was the basis for determining the number of plots, the spacing between plots, the radius of the circle (1.5 m). If there is a rationale for the survey designs, this information should also be presented.

The significance of the micro-habitat scale analysis was somewhat confusing for me. Shouldn't these be analyzed separately for each species?

Validity of the findings

Information on their ecology is useful for the conservation of two spider species. And this approach can also be used as a reference in many different systems, as habitat differences between closely related species are common to many organisms, not just spiders.

Reviewer 3 ·

Basic reporting

This study is relevant to better understand the environmental requirements of arthropods in endangered freshwater ecosystems, particularly concerning the physical (e.g., water type and speed) and architectural requirements of semi-aquatic spiders. It also contributes to understand why some species are rare, while other species, even when taxonomically close, are abundant.

The knowledge of Dolomedes populations in Europe has been fragmentary, with much anecdotal information about why two closely related species, as D. fimbriatus and D. plantarius, have such distinct distributions and relative abundances. The fact these two species, due to their morphological similarity, are frequently mistaken by each other also contributes to the lack of understanding about their respective population occurrences and how the same is related to habitat/environmental change. Therefore, a systematic study involving populations of both species unequivocally (what the authors accomplished by including the data of only adult individuals identified by the genitalia) and their respective habitats and micro-habitats was really necessary. Furthermore, the authors did a good job by sampling 31 sites and surveying many potential habitat/microhabitat traits which could potentially affect the presence of Dolomedes, which bought more precise information about the only two Dolomedes species of Europe.

However, I think the authors should also take into consideration, at least in the interpretation and discussion of their results, the importance of prey availability together with other environmental traits, even so, because many physical traits (such as environment heterogeneity/architecture/structure) tend to be positively correlated to prey abundance (e.g., Denno et al. 2005, Halaj et al. 2000). Therefore, the effect of physical variables could be confounded with the ones from prey availability. For instance, Dabrowska-Prot et al. (1968) and Luczak (1970) showed dependence between the movement and placement of Dolomedes fimbriatus individuals and prey density in a controlled field experiment. If D. fimbriatus individuals tend to displace toward locations with higher prey density, habitats with higher prey density will also tend to show high Dolomedes density. Therefore, it is possible that prey abundance could, together with architectural traits, play role in Dolomedes abundance distribution.
Differences in dispersal ability between D. fimbriatus and D. plantarius can also be taken into consideration when discussing the results. As suggested by Duffey (2012), D. fimbriatus seems a better candidate for successful terrestrial dispersal than D. plantarius, which is expected to be more dependent on water for dispersal. So, a smaller dispersal range combined with narrow habitat requirements may help to explain the more restricted distribution of D. plantarius.

The authors used elegant and flexible approaches to analyze their data (e.g., GAM, machine learning, Flexible Discriminant Analysis), which is one of the high points of this paper. This allowed to model the relationship between spider abundance and habitat traits more realistically, beyond a linear relationship. For instance, the FDA has been successful in concentrating around 91% of the total data variability in the two first axes, which makes it easy to see which environmental variables are the most related to the presence/absence of D. fimbriatus and D. plantarius.

However, I think the results should be better presented. Regarding Flexible Discriminant Analysis (FDA), I have the following doubts:
(1) Does the Flexible Discriminant Analysis deal well with non-continuous, discrete predictors or with non-ordinal/pure categorical ones, such as forest type?
(2) It has been a time since I last run a discriminant analysis (and it was a Linear instead of the Flexible). However, I remember that, unlike many ordination methods (such as PCA, RDA, CA), which are mainly exploratory, discriminant analyses are used in hypothesis testing in order to state how reliable are the group separation. Where are the results of the FDA's formal hypothesis test?
(3) I think a biplot (i.e., with the scores from both spider occurrence and habitat traits) of the FDA's two first axes would be more informative, so it will be possible to graphically visualize the patterns relating the species presence/absence with specific levels of the environmental variables.
(4) I also missed, in the main text, more information about the direction of the patterns. For instance, from lines 208 to 213 (in Results), it is just stated that water type and surrounded forest best discriminates the groups at the first axis. However, water type has four levels (creek, lave, rives, other) and surrounding forest has seven (deciduous, fields, infrastructure, pine, pine deciduous, spruce, other) and there is no straightforward information (in the main text) about what of these specific levels are positively (or negatively) related to the presence of D. fimbriatus and/or D. plantarius. This makes the reader go back and forth between Figure 3 and Table 4 to assess the patterns of the main results.
(5) The lack of the FDA biplot and clear information (in the main text) about the correlation direction between the presence/absence of D. fimbriatus/D. plantarius and the levels of each environmental variable in Results turn the same somewhat disconnected with the Discussion. For instance, the data/results patterns in the first paragraph of the Discussion (lines 233-248) are not clearly presented in the Results.
(6) I think basic descriptive results should also be added. Simple, quantitative information about the frequency of each species among habitat types would help the reader to assess the results of the formal hypotheses tests. For instance, at lines 241-242 in the Discussion “… We found D. plantarius most often at sites with slow-flowing rivers, and we found D. fimbriatus most often in bogs …”, this statement along with the FDA results, would be better supported with a basic description of species frequency among the categories of water types and speed.

References:
Dabrowska-Prot E, Luczak J, Tarwid K. 1968. Prey and predator density and their reactions in the process of mosquito reduction by spiders in field experiments. Ekologia polska 16(40): 773-819.
Denno RF, Finke DL, Langellotto GA. 2005. Direct and indirect effects of vegetation structure and habitat complexity on predator-prey and predator-predator interactions. In Barbosa P, Castellanos I, ed. Ecology of predator-prey interactions. New York: Oxford University Press, 211-239.
Halaj J, Ross DW, Moldenke AR. 2000. Importance of habitat structure to the arthropod food‐web in Douglas‐fir canopies. Oikos 90(1): 139-152.
Luczak J. 1970. Behaviour of spider populations in the presence of mosquitoes. Ekologia polska 18(31): 625-634.

Experimental design

I would like to know whether differences in reproductive periods and life cycles between D. plantarius and D. fimbriatus would interfere with the final results. This is because the authors based the data majorly on adult reproductive females. So, the presence/absence of a given species in the author’s dataset will also depend on whether both species reproduce simultaneously. If, for instance, D. plantarius tends to reproduce earlier (or later) in relation to D. fimbriatus this would ultimately lead to a bias in filed counting. According to data provided by Spider Recording Scheme and the British Arachnological Society, in Britain, adults of D. plantarius are found from late spring to late summer (~ May to August) while the ones of D. fimbriatus are found throughout spring and in early summer (~ March to June). According to Smith (2000), most of D. plantarius nursey webs are built in late July and early August; Arnqvist (1992) collected adult males and females of D. fimbriatus in June; Lecigne (2016) reported an abundance peak of D. plantarius adults in June/July and suggested that the co-existence of D. fimbriatus and D. plantarius in the same site is possible due to niche partitioning. A significant overlap of reproductive periods and phenology between the two species is necessary to compare the abundance of the two adult populations assessed simultaneously. I couldn’t find in the text information about when the authors collected the data. Additionally, have the authors included adult males in the data set? If no, why?

The authors certainly did an amazing job by sampling 31 different sites. However, I am worried about the possibility of some degree of dependence among sites, as closer water bodies tend to be similar to each other. How have the authors dealt with the possibility of spatial dependence among sites?
At lines 176-178 in Materials & Methods “… In addition, we used a single-season occupancy model (MacKenzie et al., 2002) within the unmarked package (Fiske & Chandler, 2011) to predict the nursery detection probability pooled for both species …”; why was the data of D. fimbriatus and D. plantarius pooled together, as it is possible that each species has different micro-habitat preferences for the construction of the nursery-web?

References:
Arnqvist G. 1992. Courtship behavior and sexual cannibalism in the semi-aquatic fishing spider, Dolomedes fimbriatus (Clerck) (Araneae: Pisauridae). Journal of Arachnology 20(3): 222-226.
Lecigne S. 2016. Redécouverte de Dolomedes plantarius (Clerck, 1758) (Araneae, Pisauridae) en région Nord-Pas-de-Calais (France), actualisation de sa distribution en France et aperçu de la situation en Europe. Revue arachnologique 2(3): 28-41.
Smith H. 2000. The status and conservation of the fen raft spider (Dolomedes plantarius) at Redgrave and Lopham Fen National Nature Reserve, England. Biological Conservation 95(2): 153-164.

Validity of the findings

The paper’s findings are valid to better understand the species-specific requirements of semi-aquatic spiders in freshwater environments, mainly concerning physical aspects of the habitat, such as architectural plant traits and water features (e.g., type, pH, speed, clearness).
Spiders’ populations are affected by biotic factors, such and prey abundance, dispersal rate, and predator pressure; and, as I said above, these factors should be pondered when interpreting this study's findings.

However, the populations of semiaquatic spiders are particularly affected by abiotic factors due to their dependence on water bodies. So, understanding how factors such as water pH, speed of water flow, humidity, the physical arrangement of the branches (i.e., architecture) from riparian vegetation, and water quality, in general, affect the presence of such spiders is of major relevance to restore and conserve their populations.

The paper also points out how fragmented observations of previous studies on natural history, such as the relative scarcity and apparent narrower habitat preferences of D. plantarius could be applied to broader questions in conservation.

Additional comments

At lines 30-32 of the abstract, the clauses “… We found that habitat requirements were narrower for D. plantarius compared to D. fimbriatus …” and “… the detection of nursery webs can be affected by weather conditions …” seem disjointed within the same sentence.

At lines 44-45 in Introduction, “… the first world’s scientists warning to humanity …”; I think to use of the term “first world” (as opposed to third world), if that is what the authors mean, may be outdated and inappropriate, as “first, second and third world” were terms coined in a Cold War context and they don’t reflect the relative importance of the opinions of scientists coming from these countries.

At lines 74-75 in the Introduction “… Habitat loss might have more severe consequences for D. plantarius, which has less plasticity …”. The term plasticity should be used more carefully, as it doesn’t necessarily a species with broad ecological requirements is “more plastic” than a species that requires more strict environmental conditions. For instance, a more generalist species could have their broad requirements genetically fixed through many generations of selective pressures for less habitat specificity, but it doesn’t mean it is more “plastic” in the sense that a given genotype could yield reaction norms. In order to prove plasticity, a set of specific experiments must be carried out in order to demonstrate a determined genotype can generate different phenotypes of a trait according to stimuli variation.

At lines 111-112 in Materials & Methods “… see transect description below …”; the authors should refer to Figure 2.
At lines 122-123 in Materials & Methods “… Further, we detected spiders and nursery during the systematic data collection on transects, increasing the duration of the effective search …”. Does the time the authors spent collecting the data (and consequently recorded additional spider individuals and nursery webs) was similar for each site, as the longer the time the higher the chance of recording more individuals/nursery webs? In order to keep the locations comparable, the sampling effort should be approximately the same for each site.

At lines 179-181 in Materials & Methods, “… We considered weather, microclimatic variables (wind, cloudiness, rain, shade), vegetation structure and sampling related variables as potentially influencing detectability …”; list what specifically were the variables related to weather, vegetation structure and sampling; cite the table with their respective details.

To better understand Figure 2, I depended too much on the main text (lines 141-155 of Materials & Methods). So, I think Figure 2 caption needs more development in order to be more self-explanatory and informative. Furthermore, Figure 2 pictures only the case with open water, while the authors carried out two more transect arrangements (no open water with gradient and no opens water without gradient); and the terrestrial plot located at 12m from the water is missing in the figure.

In Table 2: (1) complement the name of the first column with “Percentage of Cover” instead of just “Percentage”; (2) at the second column name “Number.of.occurrences”, replace the dots with spaces; (3) at the third column, the categories “2m”, “2a”, and “2b” were not originally from Braun-Blanquet (1928, 1932, 1964) but indeed subcategories implemented by Barkman et al. (1964) to more refinement; (4) again, I feel the Table 2 caption could be more informative, so the reader can be more independent of the main text to understand the table.

In Table 3: (1) At the third line (Aquatic vegetation) replace the name Braun-Blanquet with the acronym “BB” which is already established in table’s caption; (2) Why at the lines 148-149 of materials & methods “… Terrestrial plots were located at two, seven and twelve meters from the water edge …” the authors inform the variable plot distance from water had four levels (2, 7 and 12 meters from the water edge, plus the plot on the edge) and at the fourth line of the table (Distance to water) the level 12m is missing?; (3) At the sixth line (Dominant plant group), what were the criteria to define the groups (taxonomical, functional, architectural, …) and whether a group was dominant or not in each plot? I couldn’t find this information on the main text; (4) At the seventh line (Horizontal cover), how was this measure taken? I couldn’t find this information on the main text; (5) From lines 15th to 20th (Carex spp., Juncus spp., Typha spp., Phragmites spp., Sphagnum spp., and Deciduous plants), the authors expressed the Braun-Blanquet scale as abundance, what I understand as the number of plant individuals; whereas in Table 2 the Braun-Blanquet scale was defined as a combination of abundance and percentage of cover. In their book chapter, Westhoff & Van Der Maarel (1978) named the estimation as “cover-abundance”. Because some plant individuals can be broad and others thin, cover and abundance are not always equivalent. Furthermore, the abbreviation “spp.” should not be italicized.

In Table 4, it would be more interesting to list the variable levels in descending order according to their load values. This would allow the reader an easier assessment of the variables which most contribute to group discrimination.

In Figure 3, it seems the shape that represents sites with both species (observed) is circle and not square, as stated in the caption. I also think the captions need some rewritten for more clarity. For instance, at first glance, it seems triangle AND green define sites with only D. fimbriatus, and the information that shape defines actual observation and colour expected observations only appear at the bottom. Maybe a better it could appear like this: “Plot of the results from the flexible discriminant analysis (FDA), where colours represent observed occurrences of the species, shapes predicted occurrences, and ellipses indicate uncertainty of predicted species occurrences (95% confidence intervals). Presence of both species (observed: red, actual: circle), only D. fimbriatus (observed: green, actual: triangle) …”. Furthermore, I wonder whether a biplot (with both scores, for sites and the environmental variables) would be a more interesting option, as it would also show how the sites are grouped around some of the predictors. A biplot would also work better with Table 4.

At lines 218-219 in Results “… humidity at ground level (selection frequency = 0.02) and the intercept (selection frequency = 0.01) …”, the selection frequency of the intercept is not consistent with Figure 4, as in the latter intercept is showed as the second most important, just behind “Distance water”. Additionally, I think some explanation about what “intercept” means, in this case, would help.

In Figure 4, the caption writing does not match the plot. For instance, the plot variables “Distance water”, “Sedges”, “Crossing structure”, “Site ID”, and “Humidity ground” are captioned as “distanceWater”, “rCarex”, “rCrossingStr”, “siteID”, and “humidityGround.sc”, respectively. Furthermore, an explanation of "intercept" (in the plot) is also missing in the caption.
In Figure 5 plot A, the labels of the categories “no Water” and “other” are overlaid. Furthermore, the meaning of the category “other” should be explained in the caption.

At lines 252-253 “… This observation is validated for one of the sympatric sites sampled (Fig. 1) …” and “… we found two sympatric populations on the vegetation at the shore of a lake (Finjasjön lake, Fig. 1) …” in the Discussion, the citation of Figure 1 would be justified if this specific fen and the Finjasjön lake were identified on the map.

References:
Barkman JJ, Doing H, Segal S. 1964. Kritische bemerkungen und vorschläge zur quantitativen vegetationsanalyse. Acta botanica neerlandica 13(3): 394-419.
Braun-Blanquet, J. 1928. Pflanzensoziologie: grundzüge der vegetationskunde. Berlin Heidelberg: Springer-Verlag, pp. 37-52, 197.
Braun-Blanquet, J. 1932. Plant sociology: the study of plant communities. New York: McGraw-Hill, pp 30-39, 53.
Braun-Blanquet, J. 1964. Pflanzensoziologie_ Grundzüge der Vegetationskunde. Wien: Springer-Verlag, pp. 32-41, 52-53.
Westhoff V, Van Der Maarel E. 1978. The braun-blanquet approach. In: Classification of plant communities. Springer, 287–399.

---

## Round 0.2 · accepted · Accept

Dear Dr. Dickel,

After a new review round, two original reviewers recommended the acceptance of your manuscript. Congratulations and Merry Christmas!

·

Basic reporting

The reporting is clear, professional, and easy-to-understand. The authors have now addressed all of my concerns with their revisions.
I did catch one very small error. On the caption for figure 5, there should be a comma before "(E)", not a period.
The authors provide more introductory background now with their revision, as requested. The figures look better now that they have conducted their revisions. Results are relevant to their study and their hypotheses.

Experimental design

The research is original primary research and does fit within the Aims and Scope of PeerJ.
The research question is well defined, relevant, and meaningful, and it is state how the research fills an identified knowledge gap.
The investigation has limitations, though the authors admit these limitations in their discussion. Specifically, they admit that their sample size is small and that their sampling was limited, preventing them from making some more advanced conclusions.
The revisions that were completed improved the clarity of the methods.

Validity of the findings

Findings are novel and add to our knowledge of Dolomedes habitat preferences within Fennoscandia. Conclusions are related back to the conservation status of one of the two species studied.
Data have been provided and seem sound.
Results and conclusions are well stated, especially after the revisions.

Additional comments

Great work!

Reviewer 2 ·

Basic reporting

I thought the English expression and logic could have been further improved by revisions in response to several reviewers' comments.

Experimental design

What I did not understand about the survey design was resolved by the author's comments.

Validity of the findings

As with the previous comments, I think this study has provided valuable insights into the conservation of wetland organisms.

Additional comments

None in particular.